# Gut microbiome compositional clusters in association with cardiovascular risk: An observational cohort study

Negin Mahmoudi Hamidabad[1☉], Matteo Manzato[1], Takumi Toya[1,2☉], Lilach O. Lerman[3☉], Amir Lerman[1☉*]

1 Department of Cardiovascular Medicine, Mayo Clinic, Rochester, Minnesota, United States of America, 2 Department of Cardiology, National Defense Medical College, Saitama, Japan, 3 Division of Nephrology and Hypertension, Department of Medicine Mayo Clinic, Rochester, Minnesota, United States of America

☉ These authors contributed equally to this work.
* lerman.amir@mayo.edu

## Abstract

### Aims

The gut microbiome (GM) is increasingly recognized for its role in atherosclerosis development. However, its potential as a biomarker for risk-stratification in patients with atherosclerotic cardiovascular (CV) comorbidities remains under-explored. This study aimed to identify distinct GM clusters associated with elevated CV risk.

### Methods

In this prospective observational cohort, patients with coronary artery disease, hypertension, hyperlipidemia, or diabetes mellitus referring to Mayo Clinic from 2013 to 2018 were enrolled. Bacterial DNA was analyzed in the V3-V5 region of 16S rDNA. Beta-diversity was plotted using Principal Coordinates Analysis. Unsupervised hierarchical clustering of the GM classified participants into two clusters. Cox regression evaluated the association between clusters and Major Adverse Cardiac Events (MACE), defined as a composite of cardiac events, heart failure, and all-cause mortality. Permutational Multivariate Analysis of Variance identified clinical factors contributing to cluster assignment. Linear Discriminant analysis identified GM taxa with differential abundance among clusters and their effect sizes.

### Results

Among 211 participants (median age 60 [IQR: 50–70] years; 57.3% male), two distinct GM profiles emerged (Cluster H: N = 104; Cluster L: N = 107, P < 0.001). Cluster L participants were younger (P < 0.001), more likely female (P = 0.009), and had healthier CV profiles, including lower BMI (P = 0.007), hypertension (P = 0.010), hyperlipidemia (P = 0.005), and lower coronary artery disease prevalence (P = 0.003). Over a

**Data availability statement:** There are ethical and legal restrictions which prevent the public sharing of minimal data for this study. The data contain potentially identifiable participant information, and public sharing may breach ethical agreements. Data for this study are available upon request from Sue Graff, Operations Manager at Mayo Clinic, via email (graff.susan@mayo.edu), for researchers who meet the criteria for access to confidential information.

**Funding:** The author(s) received no specific funding for this work.

**Competing interests:** The authors have declared that no competing interests exist.

median follow-up of 7.4 years, Cluster L had a significantly lower incidence of MACE compared to Cluster H (HR = 0.48, 95% CI: 0.26–0.91, P = 0.024). Cluster L had higher operational taxonomic units (P < 0.001) and lower Bacillota-to-Bacteroidetes ratio (P < 0.001) compared to Cluster H. The predominant taxa in Cluster L included *Bacteroides*, *Alistipes*, and *Parabacteroides*, whereas *Blautia*, *Agathobacter*, and *Clostridium sensu stricto-1* were more abundant in Cluster H.

## Conclusion

Distinct GM profiles are associated with varying CV risk, highlighting the potential of unsupervised GM profiling as a novel tool for risk stratification and individualized therapy.

---

## Introduction

The human gut microbiome (GM) comprises a vast community of microorganisms that modulate host physiology, metabolism, and immune function [1]. Atherosclerosis, a leading cause of mortality and morbidity worldwide, is profoundly impacted by the GM through the production of active metabolites such as short-chain fatty acids, trimethylamine N-oxide (TMAO) and secondary bile acids, and by regulating inflammation, immune system and metabolic pathways [2,3]. Alterations in GM composition have been observed in various cardiovascular (CV) comorbidities associated with atherosclerosis, including hyperlipidemia, hypertension, diabetes mellitus, and obesity [2,4]. Furthermore, coronary artery disease (CAD) has been linked to GM compositional alterations [2,5]. Our previous research demonstrated specific GM differences in patients with advanced CAD compared to those without significant CAD, including increased *Ruminococcus gnavus* and decreased *Lachnospiraceae NK4B4 group* and *Ruminococcus gauvreauii* [5].

However, disentangling the independent associations between individual CV comorbidities and GM composition is challenging due to their frequent co-occurrence and overlapping pathophysiology, as well as the inherent complexity of the microbiome [6]. Moreover, the direct relationship between GM composition and CV outcomes remains understudied. To address these limitations, previous studies have employed clustering methods to classify patients based on CV risk profiles [7,8]. For example, Prins et al. used supervised clustering of CV risk factors based on Framingham risk scores, identifying bacterial species from *Collinsella*, *Flavonifractor*, and *Ruthenibacterium* genera associated with higher CV risk [8]. Similarly, Wang et. al, used unsupervised clustering of metabolic parameters to define metabolic phenotypes associated with CV risk, revealing high-risk clusters characterized by hyperglycemia and obesity associated with highest CV risk and associated with differential GM composition [7].

While these studies provide valuable insights into the association between GM composition and CV risk, clustering directly based on the GM composition may reveal novel associations with CV outcomes that are independent of, or not fully captured

by, traditional risk factors. Using unsupervised GM clustering, Li et al. identified distinct clusters associated with high and low CV risk across diverse age groups [9]. In the current study, we aimed to investigate the prognostic implications of GM composition in the context of CAD and CV comorbidities. We utilized unsupervised clustering of the GM composition to explore patterns in the GM composition and their association with CV risk, without pre-defined assumptions about risk groups. Our primary objective was to identify clusters in GM composition among patients with CV comorbidities and examine their association with future CV events. Our secondary objective was to characterize the clinical features of these clusters in relation to CV comorbidities.

## Materials and methods

### Study population, clinical assessment, and outcomes

In this prospective observational cohort study, we enrolled 211 consecutive patients who underwent CV assessment between 12/01/2013 and 11/30/2018. One hundred forty-two patients out of 211 patients were patients who consented to participate and provided stool for microbiome study at the time of clinically indicated coronary angiography for the assessment of chest pain. Significant coronary artery disease (CAD) defined as more than 50% luminal stenosis was detected in 92 patients [5]. The remaining 69 patients, without known or suspected CAD based on clinical history, noninvasive stress testing, and coronary imaging studies including coronary computed tomography and/or coronary angiography, were those enrolled in a different study investigating the effects of a dietary supplement on endothelial function between 02/01/2015 and 02/28/2017 [10]. All patients underwent stool sampling for GM analysis to study the association between GM composition and CV comorbidities [4]. Exclusion criteria were a history of gastrointestinal diseases such as inflammatory bowel disease, cancer, and interventions including surgery on small or large intestines and stomach, using antibiotics or probiotics at the time of study, and a history of autoimmune diseases [5]. All patients signed informed consent before enrollment in the study. The study was approved by the Mayo Clinic Institutional Review Board and was conducted by the guidelines of the Declaration of Helsinki.

Patient demographic characteristics, past medical and medication history, and laboratory data were collected from a detailed chart review by a blinded investigator. Data were collected on the following parameters: 1) sex, age, body mass index (BMI), 2) lifestyle and dietary factors including low-fat diet (never or minimally, moderately, and strictly), current tobacco smoking, alcohol intake measured as drinking more than 3 drinks per week, and exercise defined as moderate physical activity at least two times a week all in a self-reported manner, 3) coronary vascular disease risk factors including hyperlipidemia (defined as a documented history of hyperlipidemia, treatment with lipid-lowering therapy, a low-density lipoprotein cholesterol above target (<130 mg/dL for low risk patients, <100 mg/dL for moderate-high risk patients, <70 mg/dL for very high risk, and <55 mg/dL for extreme high risk patients based on 10-year atherosclerotic cardiovascular disease risk), high density lipoprotein cholesterol <40 mg/dL in men or <50 mg/dL in women, or triglyceride >150 mg/dL, diabetes mellitus (defined as a documented history or treatment of type 2 diabetes), hypertension (defined as a documented history of or treatment for hypertension), 4) CAD, defined as a documented history of percutaneous coronary intervention or coronary artery bypass grafting for significant coronary artery stenoses, or more than 50% of luminal stenosis in any coronary arteries diagnosed by coronary angiography [5], and 5) medication history including antihypertensive, lipid lowering, and antidiabetic medications, as well as nitrates, beta-blockers, and proton pump inhibitors (PPI) taken at the time of study.

Major adverse cardiovascular events (MACE) were defined as a composite measure of cardiac events (non-fatal myocardial infarction, percutaneous coronary intervention, and coronary artery bypass grafting), heart failure (new-onset heart failure or hospitalization due to heart failure exacerbation), and all-cause mortality occurring at any time after the date of stool sample collection. Heart failure was diagnosed based on cardinal symptoms, abnormal echocardiographic findings, elevated BNP/NT-proBNP levels, cardiopulmonary exercise testing, and/or right heart catheterization, according to current heart failure guidelines [11]. Individual events were identified through a combination of institutional databases, death certificates, and detailed chart reviews, and patients with a follow-up of <30 days were excluded from the analysis. In case a patient had more than one event, the first event was considered as MACE.

## Gut microbiome assessment

All participants received stool collection kits from Fisher Scientific Inc., Pittsburgh, PA, USA. These samples were collected once at baseline during patients' stay at Mayo Clinic. In case a patient was unable to provide a sample during their stay, they were instructed to collect the sample at home and ship it through overnight mail delivery. Samples were frozen at −70 ˚C within 24 hours of receipt. The microbial DNA was extracted from the samples using the Mobio PowerSoil Kit (MoBio Laboratories, Carlsbad, CA, USA). The V3-V5 region of the 16S rDNA was sequenced, and raw 16S data were processed by IM-TORNADO to form operational taxonomic units (OTU) at a 97% similarity level [4,5]. Alpha-diversity measures were calculated based on the rarefied OTU counts. Microbial richness and evenness of the microbiome sample were assessed using the number of observed OTUs, Shannon, and Chao1 indices. Bacterial β-diversity was evaluated using the Bray-Curtis method and principal coordinate analysis (PCoA) from the R vegan package at the genus level and the results were plotted using the calculated PCoA1 and PCoA2 coordinates, representing the dissimilarity in GM composition among different patients [12]. Taxa with a total abundance present in less than 5% of patients were excluded. Silhouette graph is reported in Supplementary Figure 2 (S2 Fig). Cluster stability was assessed with Jaccard similarity index across 200 bootstrap iterations.

## Statistical analysis

Continuous and normally distributed variables are reported with mean ± standard deviation and were compared using independent samples T-test or analysis of variance. Continuous and non-normally distributed variables are described with median [25th, 75th percentiles] and were compared with Mann-Whitney U or Kruskal-Wallis tests. Categorical variables are described with frequency and percentages and were compared using the Chi-square test or Fisher's exact test. Spearman and Pearson correlations were used to compare two continuous variables or one categorical variable with one continuous variable for nonparametric and parametric variables, respectively.

The optimal number of clusters was calculated as 2 using the Silhouette method. Hierarchical clustering with Ward's minimum variance method was employed to categorize microbiome samples into 2 clusters with maximum dissimilarity in GM composition. Univariate Permutational Multivariate Analysis of Variance (PERMANOVA) was used to compare the GM distribution between the two clusters (R vegan package), and multivariable PERMANOVA was used to identify the clinical factors contributing to cluster assignment in total cohort, as well as in subgroups of patients with and without CAD [12]. Taxa present in less than 5% of the subjects were excluded from the analysis. Using the "lefser" R package, the Wilcox test and Linear Discriminant Analysis were employed to rank the taxa with differential abundance between clusters and to determine the effect size of each taxon. The effect sizes provide a measure for ranking the taxa based on their ability to discriminate between the two clusters [13]. MaAsLin2 package in R was used to identify confounding associations between microbiome and clinical covariates. The model included cluster assignment as the primary variable of interest and age, sex, BMI, alcohol consumption, dietary pattern, diabetes status, antidiabetic medication use, and PPI use as covariates.

Univariable and multivariable Cox regression analyses were used to assess the predictiveness of clusters and CV risk factors on MACE incidence over the long-term follow-up. MACE-free survival was depicted using Kaplan Meier curves and the curves were compared using the Log-rank test. Statistical analysis was performed using R version 4.3.2 and SPSS version 28. A p-value less than 0.05 was considered significant.

## Results

### Baseline characteristics

Of a total of 211 study participants the median age was 60 [IQR: 50,70] years, and 121 (57.3%) were males. High-risk cluster (Cluster H) consisted of 104 (49.3%) individuals and the remaining 107 (50.7%) were categorized into low-risk cluster (Cluster L). Bacterial beta-diversity at genus level was significantly different between the two clusters (P < 0.001)

(Fig 1). Patients in Cluster L were younger (P<0.001), more likely to be females (P=0.009) and to exercise regularly (P<0.001). There was no significant difference in the prevalence of white race between the clusters (P=0.514). Cluster L had a healthier profile regarding the established atherosclerotic CV disease risk factors, including lower BMI (P=0.007), hypertension (P=0.010) and hyperlipidemia (P=0.005), and had a lower prevalence of CAD (P=0.003). However, there was no significant difference in the prevalence of diabetes mellitus (P=0.063), congestive heart failure (P=0.106), atrial fibrillation (P=0.194), smoking (P=0.446), and alcohol consumption (P=0.134) between the clusters. There was a higher prevalence of antiplatelet (P=0.001), lipid-lowering (P=0.046), and antihypertensive medications (P<0.001) use in Cluster H, as well as a higher intake of beta-blockers (P<0.001), and proton pump inhibitors (P=0.014) (Table 1). The two

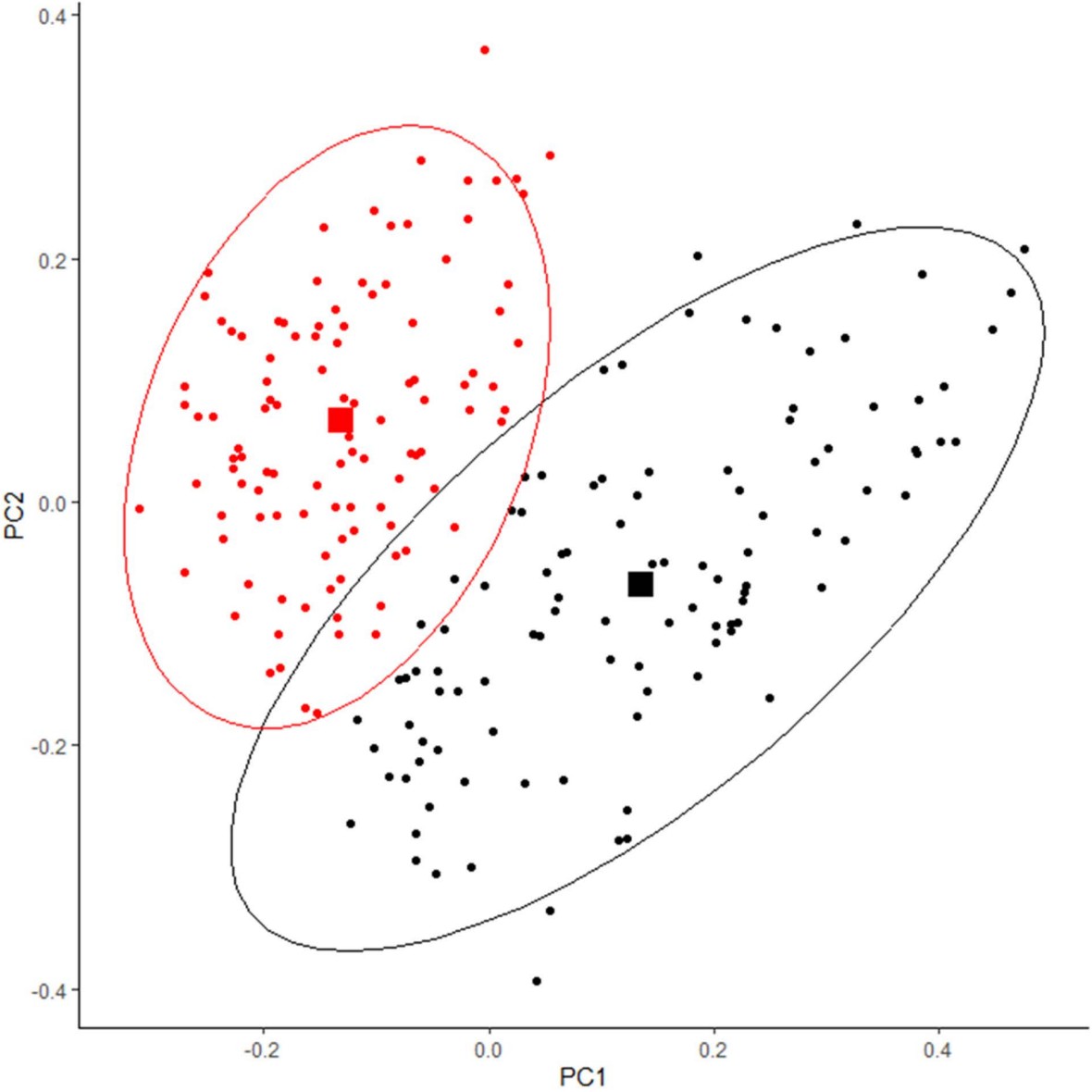

**Fig 1. Principal Coordinates Analysis (PCoA) plot illustrating beta diversity in Clusters H and L.** Significant differences in microbiome composition were observed among clusters (PERMANOVA, P<0.001). Cluster H is depicted in black, whereas Cluster L is depicted in red.

**Table 1. Baseline Characteristics.**

| | Overall N = 211 | Cluster H N = 104 | Cluster L N = 107 | P-value |
|---|---|---|---|---|
| Age, years | 60 [50-70] | 63 [56-70] | 54 [38-70] | <0.001 |
| Male sex, n(%) | 121 (57.3) | 69 (66.3) | 52 (48.6) | 0.009 |
| White race, n(%) | 206 (97.6) | 103 (99.0) | 103 (96.3) | 0.514 |
| BMI, Kg/m2 | 29.0±6.1 | 30.2±6.0 | 27.9±5.9 | 0.007 |
| Alcohol >3 drinks/week, n(%) | 39 (18.5) | 15 (14.4) | 24 (22.4) | 0.134 |
| Current smoking, n(%) | 7 (3.3) | 2 (1.9) | 5 (4.7) | 0.446 |
| Exercise, n(%) | 128 (60.7) | 51 (49) | 77 (72) | <0.001 |
| Low-fat Diet | | | | |
| Minimally | 56 (26.5) | 22 (21.2) | 27 (25.2) | 0.020 |
| Moderately | 128 (60.1) | 70 (67.3) | 58 (54.2) | |
| Strictly | 33 (15.6) | 11 (10.6) | 22 (20.6) | |
| **Cardiovascular comorbidities** | | | | |
| History of coronary artery disease, n(%) | 92 (43.6) | 56 (53.8) | 36 (33.6) | 0.003 |
| Hypertension, n(%) | 113 (53.6) | 65 (62.5) | 48 (44.9) | 0.010 |
| Hyperlipidemia, n(%) | 143 (67.8) | 80 (76.9) | 63 (58.9) | 0.005 |
| Diabetes mellitus, n(%) | 40 (19) | 25 (24) | 15 (14) | 0.063 |
| Congestive heart failure, n(%) | 23 (10.9) | 15 (14.4) | 8 (7.5) | 0.106 |
| Atrial fibrillation, n(%) | 28 (13.3) | 17 (16.3) | 11 (10.3) | 0.194 |
| **Medications** | | | | |
| Antiplatelets, n(%) | 123 (58.3) | 72 (69.2) | 51 (47.7) | 0.001 |
| Anticoagulants, n(%) | 15 (7.1) | 8 (7.7) | 7 (6.5) | 0.745 |
| Lipid Lowering medications, n(%) | 105 (49.8) | 59 (56.7) | 46 (43.0) | 0.046 |
| Antihypertensive, n(%) | 118 (55.9) | 72 (69.2) | 46 (43.0) | <0.001 |
| Antidiabetics, n(%) | 28 (13.3) | 15 (14.4) | 13 (12.1) | 0.626 |
| Nitrates, n(%) | 57 (27.0) | 33 (31.7) | 24 (22.4) | 0.128 |
| Beta-blockers, n(%) | 82 (38.9) | 55 (52.9) | 27 (25.2) | <0.001 |
| Proton pump inhibitors, n(%) | 37 (17.5) | 25 (24.0) | 12 (11.2) | 0.014 |

clusters showed good stability across the 200 bootstrap iterations. Mean Jaccard similarity was 0.767 for cluster H and 0.796 for cluster L.

## Gut microbiome clusters and study outcomes

A total of 203 patients were included in the outcome analysis, while 8 patients (6 from Cluster H and 2 from Cluster L) were excluded due to insufficient follow-up (<30 days). Over a median follow-up of 7.42 [6.04, 8.80] years, 33 (33.67%) patients in Cluster H and 14 (13.33%) patients in Cluster L experienced MACE (Cluster H: 11 heart failures, 15 cardiac events, and 7 mortalities; Cluster L: 8 heart failures, 4 cardiac events, and 2 mortalities). A total of 27 heart failures (Cluster H: 17, Cluster L: 10), 21 cardiac events (Cluster H: 16, Cluster L: 5), and 15 mortalities (Cluster H: 11, Cluster L: 4) occurred in the study population over the follow-up period. Cluster H had a higher incidence of cardiac events (P = 0.006), mortality (P = 0.039), and overall MACE (P < 0.001) compared to Cluster L. Although patients in Cluster H had a higher incidence of heart failure compared to Cluster L, the difference did not reach statistical significance (P = 0.076) (Fig 2A). In univariable Cox regression analysis, Cluster L was associated with a better MACE-free survival compared to Cluster H over the long-term follow-up (P < 0.001, HR = 0.35, 95%CI: 0.19–0.65) (Fig 2B).

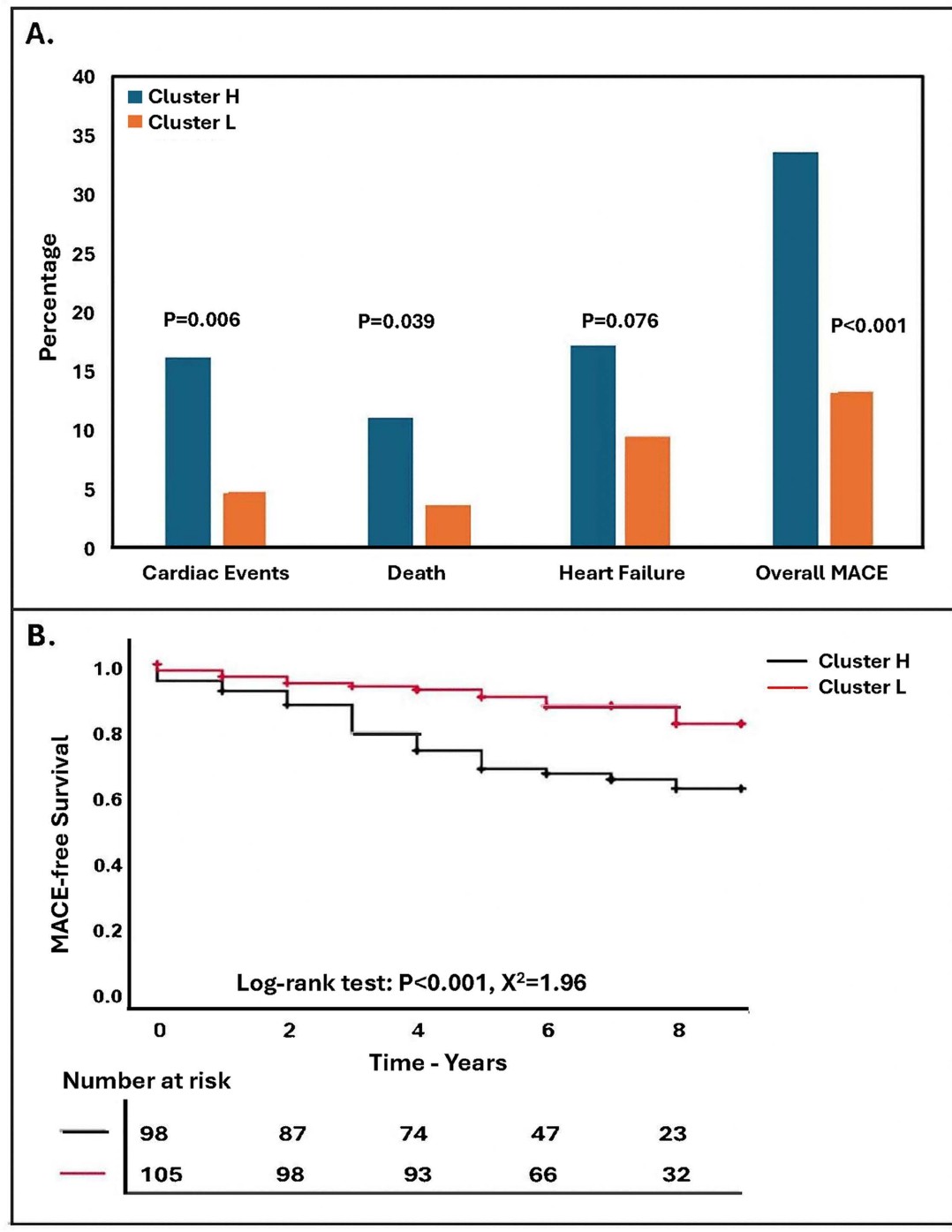

**Fig 2. Incidence of Major Adverse Cardiac Events During Study Follow-Up. A.** Incidence of Major Adverse Cardiac Events (MACE) in gut micro-biome clusters: Cluster H exhibited a significantly higher incidence of cardiac events, mortality, and overall MACE compared to Cluster L (all P<0.05). Although Cluster H had a higher rate of heart failure compared to Cluster L, the difference did not reach statistical significance (P=0.076). **B.** Kaplan-Meier curve depicting MACE-free survival in each cluster during follow-up: MACE-free survival was higher in Cluster L compared to Cluster H (P<0.001). MACE: Major Adverse Cardiovascular Events.

In multivariable Cox regression analysis, Cluster L had a better MACE-free survival compared to Cluster H, after adjusting for age, BMI, CAD, hyperlipidemia, and hypertension (P = 0.024, HR = 0.48, 95%CI: 0.26–0.91). Additionally, there was no significant interaction between the clusters and either age or CAD in predicting MACE (P = 0.101, and P = 0.117, respectively) (S1 Table). In a subgroup of patients with CAD, clusters were not predictive of MACE in univariate Cox regression analysis (P = 0.197, HR = 0.64, 95%CI: 0.32–1.27), while in the subgroup of patients without CAD, Cluster L patients had a significantly higher probability of MACE-free survival compared to Cluster H (P = 0.017, HR = 0.151, 95%CI: 0.03–0.71).

## Association between clinical factors and microbiome clusters

To understand the interplay between CV risk factors and microbial composition, and their role in differentiating between clusters, we used multivariable PERMANOVA analysis. In the overall cohort, Age ($R^2$: 0.025, P < 0.001), BMI ($R^2$: 0.023, P < 0.001), exercise ($R^2$: 0.008, P = 0.039), antidiabetic medications ($R^2$: 0.010, P = 0.009), beta-blockers ($R^2$: 0.011, P = 0.003), and proton pump inhibitors ($R^2$: 0.008, P = 0.029) contributed to the cluster distinction. However, in patients with CAD, only BMI ($R^2$: 0.023, P = 0.207) and beta-blockers ($R^2$: 0.021, P = 0.016) were significant contributors to cluster distinction. In patients without CAD, age ($R^2$: 0.034, P < 0.001) and BMI ($R^2$: 0.030, P < 0.001) were the most prominent contributors to cluster distinction similar to the overall cohort, followed by gender ($R^2$: 0.015, P = 0.015), exercise ($R^2$: 0.014, P = 0.028), and antidiabetic medications ($R^2$: 0.017, P = 0.014) (S2 Table).

Furthermore, age was inversely correlated with number of OTUs (P < 0.001, r = −0.339; S1A Fig), Shannon index (P < 0.001, r = −0.247; S1B Fig), and Chao1 index (P < 0.001, r = −0.320) and positively correlated with *Bacillota* to *Bacteroidetes* ratio (P < 0.001, r = 0.248; S1C Fig). However, after controlling for BMI, the correlation between age and Bacillotato Bacteroidetes ratio lost significance (P = 0.317).

Similarly, BMI was inversely correlated to the number of OTUs (P < 0.001, r = −0.386; S1D Fig), Shannon index (P < 0.001, r = −0.329; S1E Fig), Chao1 index (P < 0.001, r = −0.404), and positively correlated to *Bacillota* to *Bacteroidetes* ratio (P = 0.003, r = 0.205; S1F Fig).

After MaAsLin2 analysis we found significant associations between different OTUs and clinical variables. (S3 Table) A limited number of taxa within the *Bacteroid* and *Clostridiales* families showed a significant negative association with age. Within the *Clostridiales* family different taxa were negatively associated to BMI too. In contrast, alcohol use, diabetes, low-fat diet and use of PPI influenced only in small part different OTUs. Importantly, cluster assignment was significantly associated with multiple taxa from the clostridial and bacteroid families, independent of clinical variables.

## Gut microbiome composition and diversity across clusters

*Bacillota* was the most abundant phylum in both clusters. *Bacillota* and *Actinobacteria* were more abundant in Cluster H (P < 0.001 and P = 0.002, respectively), while *Bacteroidetes* and *Proteobacteria* were more abundant in Cluster L (P < 0.001 and P = 0.013, respectively; Fig 3A). There was no significant difference in TMAO between clusters (P = 0.581, Fig 3B). Subjects in Cluster L had significantly higher OTUs (P < 0.001), as well as higher Chao1 (P < 0.001) and Shannon indices (P = 0.002) (Fig 3C-E), and significantly lower *Bacillota* to *Bacteroidetes* ratio compared to Cluster H (P < 0.001). Bacterial taxa with significantly different abundances between Cluster H and Cluster L as well as the effect sizes for each taxon are represented in Fig 4. The top 3 taxa favoring Cluster L were *Bacteroides*, *Alistipes*, and *Parabacteroides*, and the top 3 taxa favoring Cluster H were *Blautia*, *Agathobacter*, and *Clostridium sensu stricto-1*.

## Discussion

The current study identified two distinct GM clusters. Patients in Cluster H were older, predominantly male, and had a higher prevalence of CAD, hypertension, and hyperlipidemia, along with a higher MACE incidence in the follow-up compared to Cluster L. In contrast, patients in Cluster L had lower BMI, exercised more, had a better GM diversity

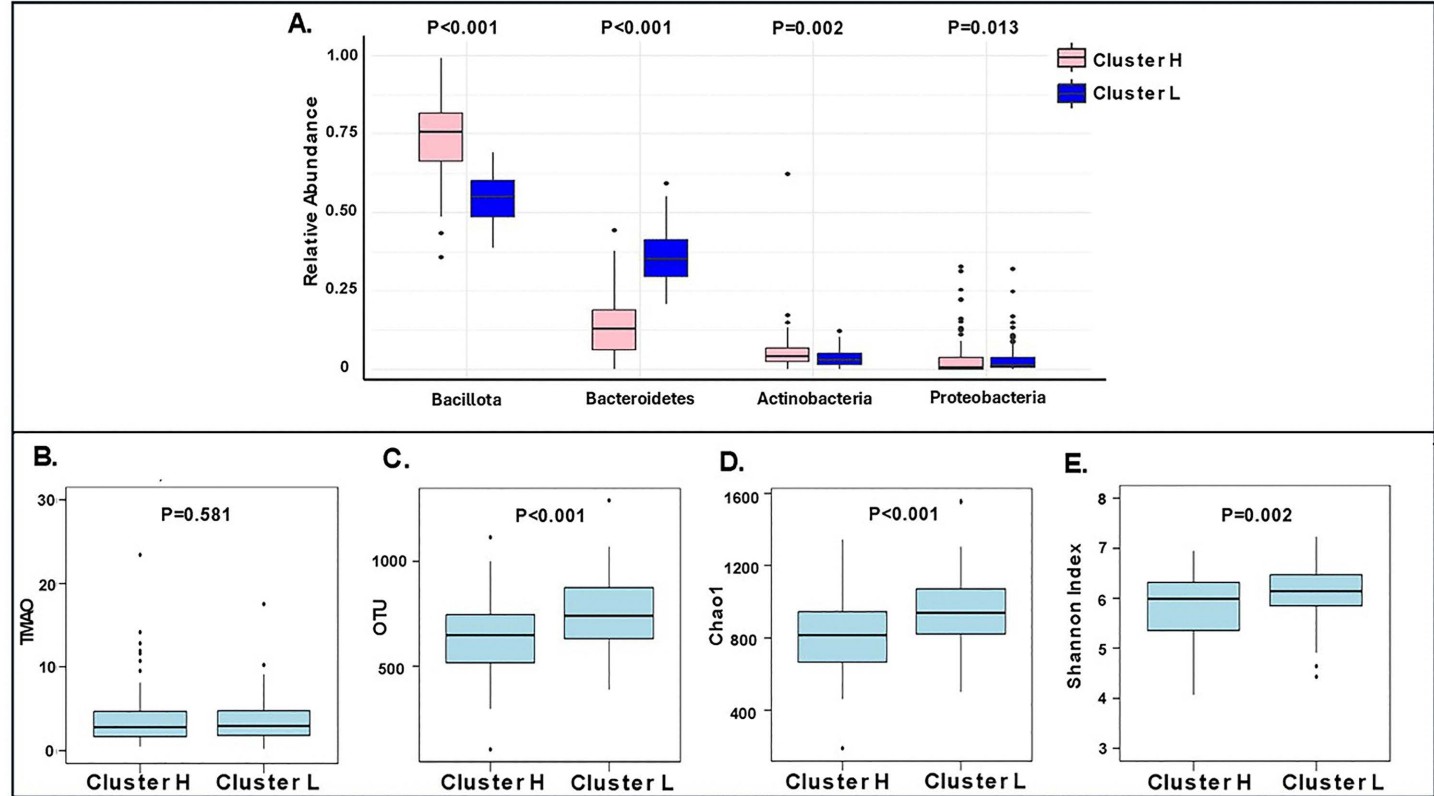

**Fig 3. Gut Microbiome Composition Across Phyla and alpha-Diversity Measures Among Clusters.** A: Box plot displaying the relative abundances of the top 4 abundant phyla across clusters. Cluster H exhibited higher abundances of *Bacillota* and *Actinobacteria*, whereas Cluster L showed higher abundances of *Bacteroidetes* and *Proteobacteria*. B: Box plot comparing trimethylamine N-oxide (TMAO) levels between clusters. No significant difference in TMAO levels was observed between clusters. C: Box plot comparing operational taxonomic units (OTUs) between clusters. Cluster L displayed a higher mean number of OTUs compared to Cluster H. D: Box plot representing the Chao1 index across clusters. Cluster L demonstrated a significantly higher Chao1 index compared to Cluster H. E: Box plot depicting the Shannon index across clusters. Cluster L exhibited a significantly higher Shannon index compared to Cluster H.

profile, and experienced less events during follow-up. Age and BMI were the most significant contributors to the cluster distinction and were independently associated with some strains, followed by beta-blocker use, antidiabetic medications, exercise, proton pump inhibitors, and gender. However, these factors accounted for only a small proportion of the variance in GM composition, reinforcing this independence and suggesting potential novel therapeutic targets for risk management.

## Gut microbiome and cardiovascular risk prediction

The GM and its metabolites play crucial roles in modulating the development and progression of atherosclerosis [2]. Alterations in GM diversity and metabolites can disrupt normal host-microbiome interactions, leading to a weakened intestinal barrier function, increased bacterial translocation, and dysregulated glucose and lipid metabolism [2]. These changes exacerbate inflammation, a key contributor to atherosclerosis, and are associated with several CV comorbidities including advanced age, obesity, diabetes mellitus, hypertension, hyperlipidemia, and CAD [2]. While associations between GM alterations and established CV risk factors are increasingly recognized [14–16], the independent contribution of the GM to CV risk prediction and its interplay with established CAD remains less understood.

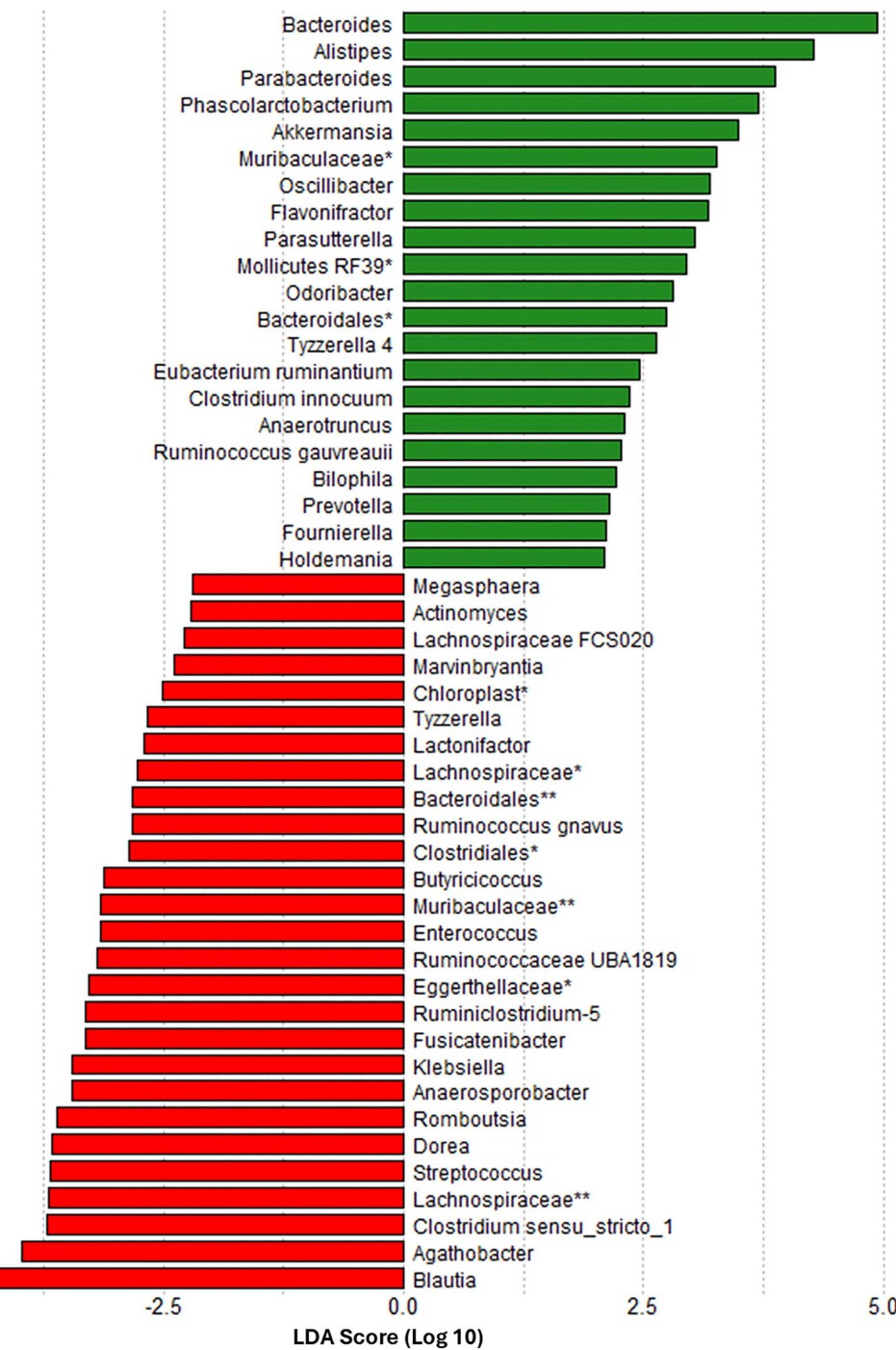

**Fig 4. Linear Discriminant Analysis (LDA) of Clinical Factors Contributing to Gut Microbiome Clustering.** Bar plot showing the LDA scores (log10) of the most influential taxa contributing to cluster separation. Higher LDA scores indicate a greater effect size, reflecting the strength of each taxon's contribution to distinguishing between clusters. Green bars: Taxa with higher abundance in Cluster L; Red bars: Taxa with higher abundance in Cluster H. * and ** indicate unclassified bacteria of a higher order.

In previous studies, such as Li et al, [9] microbiome clustering has been associated with cardiovascular risk profile and future cardiovascular risk prediction. In our study, we have correlated the differences in GM profile with incidence of MACE which provides a more comprehensive look into the actual association between GM profile and CV outcomes.

In addition, previous studies have clustered patients based on their CV risk profile, [5,17] which can bias the microbiome cluster formation. In the current study, unsupervised clustering of the microbiome can reveal hidden associations between the bacterial taxa, which may be left undiscovered when clustering is based on CV risk factors.

In this study, we identified a high-risk cluster (Cluster H) associated with an elevated risk of MACE, even after adjusting for traditional CV risk factors. This finding aligns with previous studies linking unhealthy GM profiles to higher CV events and reports of distinct GM patterns in high-risk metabolic groups, highlighting the potential of GM as an independent contributor to CV risk [7–9]. Intriguingly, while CAD patients were more likely to belong to Cluster H, the clusters themselves did not predict MACE within the CAD subgroup. While decreased statistical power due to smaller sample size may partly explain this observation it is also plausible that, in the context of established CAD, the influence of the GM on MACE is overshadowed by more dominant factors such as disease severity, pharmacologic interventions, and potentially limited influence of lifestyle factors at advanced disease stages. In contrast, among patients without established CAD, who are in earlier stages of CV risk, the microbiome may exert a more significant influence, interacting more dynamically with lifestyle factors.

## Influence of clinical factors on microbiome composition

To investigate the potential factors influencing cluster assignment, we analyzed the role of clinical factors, including established CV comorbidities. In the overall cohort, several factors were associated with cluster differentiation, including age, BMI, sex, exercise, proton pump inhibitor use, beta-blockers, and antidiabetic medications. Among these, age and BMI emerged as the most prominent contributors to cluster assignment. This finding is in line with findings of Wang et.al, who identified age and high-risk metabolic clusters, characterized by hyperglycemia and obesity, as the strongest factors correlating with GM composition. They also defined microbiome age, an index designed by 55 age-related species, where a lower microbiome age appeared to mitigate the risk associated with high-risk metabolic profiles [7]. The higher prevalence of female sex in Cluster L is consistent with previous research that women tend to have a greater GM diversity than men [18]. Additionally, physical activity can influence GM composition by decreasing colon transit time, protecting intestinal morphology and barrier integrity, and attenuating gut inflammation [19]. The associations between observed cluster assignment and the use of proton pump inhibitors, beta-blockers, and antidiabetic medications are also in line with prior findings [20]. Notably, beta-blockers have been linked to better alpha-diversity, possibly by modulating the activity of GM-derived metabolites such as phenylacetylglutamine [21,22]. Our analysis revealed distinct patterns when stratified by CAD status. In patients with CAD, whose lifestyle influences are likely attenuated by advanced disease, only BMI and beta-blocker remained significantly associated with cluster assignment. These findings underscore the importance of weight management and dietary interventions, potentially impacting events through GM modulation. The exclusive association of beta-blocker use with cluster assignment in CAD patients raises the possibility of a microbiome-mediated mechanism behind their benefits, meriting further investigation. In contrast, among non-CAD patients, a broader range of factors, including age, sex, lifestyle factors such as BMI, and exercise, and medications including proton pump inhibitors and antidiabetic medications were associated with cluster assignment. Although antidiabetic medication use was low in this group, most users were classified within Cluster L, suggesting a potentially favorable microbiome association in earlier stages of CV risk. Previous studies suggest that medications, including antidiabetic agents, can impact the GM independently of baseline disease status [20], highlighting the need for further research into how glucose modulators interact with the microbiome.

It is important to acknowledge that the contribution of measured clinical factors to cluster assignment was relatively low in both groups, consistent with findings from other studies [7]. This suggests that other unmeasured variables, such

as stochastic, genetic, and environmental influences, likely play a substantial role in shaping GM composition. While the precise factors affecting GM composition remain to be fully elucidated, targeting modifiable risk factors could significantly reduce the risk of future adverse events. These findings highlight the importance of unsupervised GM clustering in risk assessment studies, as clustering based solely on clinical risk factors may obscure inherent GM patterns not explained by traditional measures.

### Gut microbiome diversity and composition across clusters

Our analysis of the microbial composition across clusters reinforces their association with differential CV risk profiles. Consistent with our findings of better metabolic profile, lower MACE risk, and higher alpha diversity in Cluster L, this group exhibited a lower Bacillota to Bacteroidetes ratio, which is associated with obesity [23]. Additionally, Cluster L was enriched in several beneficial bacterial taxa. Notably, *Bacteroides,* inversely associated with aging, was the most prominent taxon favoring Cluster L [24]. Furthermore, Cluster L was enriched in *Alistipes* and *Parabacteroides*, associated with healthy aging [25], and *Akkermansia*, inversely correlated with obesity [26], and positively associated with healthy aging [25,27]. Conversely, Cluster H demonstrated enrichment of bacteria associated with metabolic disease such as *Blautia*, *Megasphaera*, *Fusicatenibacter*, and *Dorea [7*,9]. Furthermore, bacteria linked to CAD including *Actinomyces*, *Ruminococcus gnavus*, *Enterococcus*, *Blautia*, and *Streptococcus* were also more prevalent in Cluster H [5,28].

In conclusion, our study identified distinct GM clusters independently predictive of MACE, even after adjusting for traditional risk factors. This highlights the potential of high-risk GM cluster as a novel contributor to CV risk. Notably, the attenuated predictive power of these clusters in patients with established CAD suggests a complex interplay between disease status, medication use, and microbiome composition. These findings underscore the need for further research to elucidate mechanisms by which the GM affects events and explore microbiome-targeted therapies, particularly in the context of established CV disease. Our work suggests GM profiling as a potential risk-stratification tool and therapeutic target, laying groundwork for future individualized GM-based interventions. However several limitations must be acknowledged, including a single-center design, relatively small sample size, potential underestimation of certain comorbidities which were either unaccounted for or had low prevalence in our population, majority of individuals being white and having some baseline comorbidities, limiting the generalizability of these findings to a selected group, and lack of metabolomics data. Moreover, as the microbiome is a dynamic entity, a single assessment may not fully capture the longitudinal behavior and its interplay with CV diseases. Validation in larger, longitudinal, multi-omics studies to establish causality and translate findings into clinically actionable strategies for CV risk prediction and prevention.

### Supporting information

**S1 Table. Multivariable Cox Regression Analysis of the Predictors of the Major Adverse Cardiovascular Events Over the Long-term Follow-up.**
(DOCX)

**S2 Table. Association of Gut Microbiota with Clinical Factors.**
(DOCX)

**S3 Table. Results of MaAsLin2 analysis reporting significant associations.**
(DOCX)

**S1 Fig. Correlations of Age and Body Mass Index with Measures of Microbiome Diversity.** A. Scatterplot representing the correlation between age and OTUs. There was a moderate and inverse correlation between age and number of OTUs. B. Scatterplot illustrating the correlation between Shannon index and age. There was a gradual decrease in the

Shannon index with advancing age. C. Scatterplot depicting the correlation between age and F/B ratio. There was a weak but statistically significant correlation between age and the F/B ratio. D. Scatterplot of the correlation between OTUs and BMI. There was a moderate negative correlation between number of OTUs and BMI. E. Scatterplot of the correlation between Shannon index and BMI. There was a moderate negative correlation between BMI and Shannon index. F. Scatterplot of the correlation between BMI and F/B ratio. There was a significant correlation between BMI and F/B ratio. BMI: Body Mass Index (Kg/m$^2$), F/B ratio: Bacillota to Bacteroidetes ratio, OTU: Operational Taxonomic Units.
(TIF)

**S2 Fig. Silhouette graph.**
(TIF)

## Acknowledgments

We thank Bradley Lewis (Division of Clinical Trials and Biostatistics, Mayo Clinic, Rochester, MN) for his feedback on the methods and statistical analysis of this paper.

## Author contributions

**Conceptualization:** Negin Mahmoudi Hamidabad, Amir Lerman.

**Data curation:** Negin Mahmoudi Hamidabad, Matteo Manzato, Takumi Toya.

**Formal analysis:** Negin Mahmoudi Hamidabad, Matteo Manzato.

**Investigation:** Negin Mahmoudi Hamidabad, Amir Lerman.

**Methodology:** Negin Mahmoudi Hamidabad.

**Project administration:** Amir Lerman.

**Resources:** Amir Lerman.

**Software:** Negin Mahmoudi Hamidabad.

**Supervision:** Lilach O Lerman, Amir Lerman.

**Validation:** Negin Mahmoudi Hamidabad.

**Visualization:** Negin Mahmoudi Hamidabad.

**Writing – original draft:** Negin Mahmoudi Hamidabad.

**Writing – review & editing:** Negin Mahmoudi Hamidabad, Matteo Manzato, Takumi Toya, Lilach O Lerman, Amir Lerman.

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
