## [Decision Letter · Decision Letter 0]

3 Sep 2025

Dear Dr. Lerman,

Thank you for submitting your manuscript to PLOS ONE. After careful consideration, we feel that it has merit but does not fully meet PLOS ONE’s publication criteria as it currently stands. Therefore, we invite you to submit a revised version of the manuscript that addresses the points raised during the review process.

We look forward to receiving your revised manuscript.

Kind regards,

Satish G Patil, PhD

Academic Editor

PLOS ONE

Journal Requirements:

3. In this instance it seems there may be acceptable restrictions in place that prevent the public sharing of your minimal data. However, in line with our goal of ensuring long-term data availability to all interested researchers, PLOS’ Data Policy states that authors cannot be the sole named individuals responsible for ensuring data access (http://journals.plos.org/plosone/s/data-availability#loc-acceptable-data-sharing-methods).

Reviewers' comments:

Reviewer's Responses to Questions

**Comments to the Author**

1. Is the manuscript technically sound, and do the data support the conclusions?

Reviewer #1: Yes

Reviewer #2: Partly

2. Has the statistical analysis been performed appropriately and rigorously?

Reviewer #1: No

Reviewer #2: No

3. Have the authors made all data underlying the findings in their manuscript fully available?

Reviewer #1: Yes

Reviewer #2: Yes

4. Is the manuscript presented in an intelligible fashion and written in standard English?

Reviewer #1: Yes

Reviewer #2: Yes

Reviewer #1: I found this study intriguing and valuable in exploring the associations between gut microbiome compositional clusters and cardiovascular risk. However, I have some concerns regarding potential confounding variables. Upon examining the patient characteristics, there appear to be several significant differences between Cluster L and Cluster H, particularly in variables such as age, BMI, and other clinical parameters. These are known confounders in gut microbiome analyses and may influence the observed associations independently of microbiome composition.

To strengthen the robustness of the findings, I would suggest incorporating additional statistical methods to control for these covariates. In particular, MaAsLin2, a comprehensive multivariable association tool in R, would be highly appropriate for this kind of population-scale analysis. It allows for adjustment of multiple covariates and can help disentangle microbiome-feature associations from confounding influences. Including such an analysis could enhance the interpretability and reproducibility of the study's conclusions.

Reviewer #2: The manuscript entitled "Gut microbiome compositional clusters in association with cardiovascular risk: An

observational cohort study" addresses an important question linking gut microbiota to atherosclerotic risk. The rationale is well motivated by prior evidence that gut dysbiosis is associated with cardiovascular risk factors and disease. However, there are several issues needs to address.

1. Similar approaches have been applied to metabolic phenotypes (e.g. Li et al. 2024 found microbiome-based high/low risk clusters; 10.1016/j.lanepe.2024.101195). Discuss the major distinct findings in this study compared to prior studies

2. Figure 3 is of bad quality and the bacterial taxa name is incorrect. 'Firmicutes' is now 'Bacillota'. Please change all the bacterial taxa name with its correct name (https://lpsn.dsmz.de)

3. Microbiome data are only available upon request, which may hinder reproducibility. Public repository deposition (e.g., SRA) would improve transparency.

4. Participants were drawn from two different study settings (angiography patients and a dietary supplement trial). This introduces heterogeneity and potential selection bias. Pooling these groups without extensive stratification may confound results.

5. Gut microbiota were measured once at baseline. Since the microbiome is dynamic and influenced by diet, medications, and lifestyle changes, a single snapshot may not fully capture long term microbial patterns relevant to cardiovascular risk.

6. The Cox regression adjusted for some clinical risk factors, but sex, exercise, diet, and medication use also differed between clusters and were not fully accounted for. These could confound the observed associations.

7. Several drugs (beta-blockers, antidiabetic agents) were associated with cluster assignment. These may alter gut microbiota independently of cardiovascular risk and could drive clustering.

8. All subjects had at least one cardiovascular risk factor (CAD, hypertension, hyperlipidemia, or diabetes). While this focuses on a relevant patient population, it means there is no truly healthy control group. The authors should note that findings reflect variation within an at-risk population, which may limit generalizability. Also, the cohort is almost entirely White (∼98%); this should be stated as a limitation.

9. Hierarchical clustering (Ward’s method) using Bray–Curtis distances is reasonable. The manuscript states that the Silhouette method indicated two clusters; it would improve transparency to report the actual Silhouette score or show the clustering tree (dendrogram) in a supplement. Details on preprocessing should be explicit: for instance, how were OTU counts normalized or rarefied before clustering? Additionally, assessing cluster stability (e.g. by bootstrapping or using an alternative method like k-means) would strengthen confidence that the two-cluster solution is robust.

10. he predominant genera (e.g. higher Bacteroides/Alistipes in the lower-risk cluster, higher Blautia/Clostridium in the higher-risk cluster) are interesting. The discussion could be enriched by relating these to known physiology or prior studies (for example, some Bacteroides species are linked to leaner phenotypes). This would highlight the novelty of identifying these genera in a cardiovascular risk context.

11. The manuscript is generally well-organized, but some areas could be clearer. Define the cluster labels (‘Cluster H’ and ‘Cluster L’) at first mention to avoid confusion. Consider using descriptive terms (“high-risk” vs “low-risk”) alongside the letters.

12. Kindly Re-run the Cox models including sex and other differing baseline factors (exercise, diet category, medication use) to confirm the cluster effect is independent. Report the full multivariable results (in Supplement) to demonstrate which covariates were significant.

13. The finding that clusters predicted MACE mainly in patients without established CAD is interesting. It could be explored further or at least discussed (is it due to sample size or a real difference?). If feasible, test interaction between cluster and CAD status.

14. Please analyze taxa at the species/ASV level for the top genera. This might reveal which specific microbes drive the associations.

15. Clarify whether microbiome samples were taken once (at baseline) and how far in advance of events. If only a single timepoint is used, note this limitation as microbiome can change over time.

**Do you want your identity to be public for this peer review?** For information about this choice, including consent withdrawal, please see our Privacy Policy

Reviewer #1: **Yes:** Hajar Fauzan Ahmad

Reviewer #2: No

---

## [Author Response · Author response to Decision Letter 1]

16 Dec 2025

Reviewer #1

1. I found this study intriguing and valuable in exploring the associations between gut microbiome compositional clusters and cardiovascular risk. However, I have some concerns regarding potential confounding variables. Upon examining the patient characteristics, there appear to be several significant differences between Cluster L and Cluster H, particularly in variables such as age, BMI, and other clinical parameters. These are known confounders in gut microbiome analyses and may influence the observed associations independently of microbiome composition.

To strengthen the robustness of the findings, I would suggest incorporating additional statistical methods to control for these covariates. In particular, MaAsLin2, a comprehensive multivariable association tool in R, would be highly appropriate for this kind of population-scale analysis. It allows for adjustment of multiple covariates and can help disentangle microbiome-feature associations from confounding influences. Including such an analysis could enhance the interpretability and reproducibility of the study's conclusions.

We thank the reviewer for the insightful comment. We acknowledge that some clinical characteristics may influence gut microbiome composition. For this reason, we have conducted further analysis with the MaAsLin2 tool in R, adjusting for the following clinical covariates:

- Age

- Sex

- BMI

- Alcohol

- Diet

- Diabetes

- Antidiabetic medications

- Proton Pump inhibitors

In brief, we did find significant association between clinical variables and microbiome. Both the Bacteroid and Clostridiales families significantly decreased with increasing age, with Clostridiales also decreasing with increasing BMI. On the other hand, some Actinobacteria species were increased with alcohol use. Importantly, cluster assignment remained significantly associated with the most relevant microbial features, indicating that some microbiome traits cannot be fully explained by clinical covariates alone. We acknowledge that additional unmeasured variables may contribute to microbiome variation. Nonetheless, these results support the presence of independent microbial signatures distinguishing the two clusters. We have included in Supplementary Table 3 (S3 Table) the significant associations resulting from MaAsLin2 analysis.

We have also updated different sections of the paper to reflect these changes.

The methods section now reads:

“MaAsLin2 package in R was used to assess independent associations between taxa and clinical covariates.”

The results section now reads:

“After MaAsLin2 analysis we found significant associations between different OTUs and clinical variables. (S3 Table) A limited number of OTUs within the Bacteroid and Clostridiales families showed a significant negative association with age. Within the Clostridiales family different OTUs were negatively associated to BMI too. In contrast, alcohol use, diabetes, low-fat diet and use of PPI influenced only in small part different OTUs. Importantly, cluster assignment was significantly associated with multiple OUT from the clostridial and bacteroid family, independent of clinical variables.”

Reviewer #2

The manuscript entitled "Gut microbiome compositional clusters in association with cardiovascular risk: An observational cohort study" addresses an important question linking gut microbiota to atherosclerotic risk. The rationale is well motivated by prior evidence that gut dysbiosis is associated with cardiovascular risk factors and disease. However, there are several issues needs to address.

1. Similar approaches have been applied to metabolic phenotypes (e.g. Li et al. 2024 found microbiome-based high/low risk clusters; 10.1016/j.lanepe.2024.101195). Discuss the major distinct findings in this study compared to prior studies

We thank the reviewer for the comment. We have expanded our discussion to compare our study to the one mentioned. The following sentences have been added:

“In previous studies, such as Li et al, microbiome clustering has been associated with cardiovascular risk profile and future cardiovascular risk prediction. In our study, we have correlated the differences in GM profile with incidence of MACE which provides a more comprehensive look into the actual association between GM profile and CV outcomes.

In addition, previous studies have clustered patients based on their CV risk profile, which can bias the microbiome cluster formation. In the current study, unsupervised clustering of the microbiome can reveal hidden associations between the bacterial taxa, which may be left undiscovered when clustering is based on CV risk factors.”

2. Figure 3 is of bad quality and the bacterial taxa name is incorrect. 'Firmicutes' is now 'Bacillota'. Please change all the bacterial taxa name with its correct name (https://lpsn.dsmz.de)

We appreciate the reviewer insight. Figure 3 has now been adjusted both in quality and nomenclature. In concordance with the new nomenclature, “Firmicutes” has been replaced by “Bacillota” in the remainder of the manuscript too.

3. Microbiome data are only available upon request, which may hinder reproducibility. Public repository deposition (e.g., SRA) would improve transparency.

We thank the reviewer for the comment. We are currently working with our institutional review board to explore the possibility of sharing data, limited to the microbiome composition, in a public repository. Should IRB approve this request, we will proceed with data sharing.

4. Participants were drawn from two different study settings (angiography patients and a dietary supplement trial). This introduces heterogeneity and potential selection bias. Pooling these groups without extensive stratification may confound results.

We thank the reviewer for the insightful comments. We acknowledge that pooling participants from two different study settings may introduce selection bias. To address this, we adjusted for clinical variables across different models. In the updated Supplementary Table 1, cluster assignment predicted MACE after adjusting for the significant differences present between the two clusters observed in the demographics, lifestyle (diet and exercise) and comorbidities. As we understand that multivariable models can only partially adjust for potential selection bias, we have also updated the limitation statement as follows:

“However several limitations must be acknowledged, including a single-center design, relatively small sample size, potential underestimation of certain comorbidities which were either unaccounted for or had low prevalence in our population and lack of metabolomics data.”

Please find the updated tables in Supplementary Table 1.

5. Gut microbiota were measured once at baseline. Since the microbiome is dynamic and influenced by diet, medications, and lifestyle changes, a single snapshot may not fully capture long term microbial patterns relevant to cardiovascular risk.

We thank the reviewer for the comment. As multiple microbiome assessments were not performed, we have included this aspect in the limitation paragraph. The following sentence has been added:

“Moreover, as the microbiome is a dynamic entity, a single assessment may not fully capture the longitudinal behavior and its interplay with CV diseases.”

6. The Cox regression adjusted for some clinical risk factors, but sex, exercise, diet, and medication use also differed between clusters and were not fully accounted for. These could confound the observed associations.

We appreciate the reviewer’s observation. We have now adjusted for several clinical variables across different models. In brief, Cluster assignment remained a significant predictor of MACE after running different cox multivariable models including the factors not previously included. Please find the different cox models in the updated Supplementary Table 1.

7. Several drugs (beta-blockers, antidiabetic agents) were associated with cluster assignment. These may alter gut microbiota independently of cardiovascular risk and could drive clustering.

We appreciate the reviewer insight and we agree with the statement. For this reason, we performed further analysis with MaAsLin2 tool in R, which allows to adjust for multiple covariates and explore specific associations between clinical and microbiome variables. Concerning drugs, we have included proton pump inhibitors and antidiabetic agents in the model. We found that antidiabeticv agents were significantly associated with modest higher abundance of Gammaproteobacteria, while use of proton-pump inhibitors were significantly associated with a very modest higher abundance of Actinobacteria. However, even when factoring in other clinical variables, such as Age, Sex, BMI, Alcohol use, diet and diabetes, cluster assignment was significantly associated with certain microbial features, showing that certain traits cannot be fully explained by clinical variables alone.

We have also updated different sections of the paper to reflect these changes.

The methods section now reads:

“MaAsLin2 package in R was used to assess independent associations between taxa and clinical covariates.”

The results section now reads:

“After MaAsLin2 analysis we found significant associations between different OTUs and clinical variables. (S3 Table) A limited number of OTUs within the Bacteroid and Clostridiales families showed a significant negative association with age. Within the Clostridiales family different OTUs were negatively associated to BMI too. In contrast, alcohol use, diabetes, low-fat diet and use of PPI influenced only in small part different OTUs. Importantly, cluster assignment was significantly associated with multiple OUT from the clostridial and bacteroid family, independent of clinical variables.”

8. All subjects had at least one cardiovascular risk factor (CAD, hypertension, hyperlipidemia, or diabetes). While this focuses on a relevant patient population, it means there is no truly healthy control group. The authors should note that findings reflect variation within an at-risk population, which may limit generalizability. Also, the cohort is almost entirely White (∼98%); this should be stated as a limitation.

We thank the reviewer for the observation. The following sentence has been added to the limitation paragraph to reflect our population characteristics.

“However several limitations must be acknowledged, including […] majority of individuals being white and having some baseline comorbidities, limiting the generalizability of these findings to a selected group, […].”

9. Hierarchical clustering (Ward’s method) using Bray–Curtis distances is reasonable. The manuscript states that the Silhouette method indicated two clusters; it would improve transparency to report the actual Silhouette score or show the clustering tree (dendrogram) in a supplement. Details on preprocessing should be explicit: for instance, how were OTU counts normalized or rarefied before clustering? Additionally, assessing cluster stability (e.g. by bootstrapping or using an alternative method like k-means) would strengthen confidence that the two-cluster solution is robust.

We thank the reviewer for the comment. We have added the following sentence to the method section to clarify pre-processing:

“Taxa with a total abundance present in less than 5% of patients were excluded.”

We have now also reported the Silhouette graph as Supplementary Figure 2, showing higher score with 2 clusters. To assess the stability of clusters we used Jaccard similarity index. Mean Jaccard similarity for H and L clusters were 0.767 and 0.796 respectively, indicating higher similarity, and thus stability, across all bootstrap iterations.

The methods section has been updated as follows:

“Cluster stability was assessed with Jaccard similarity index across 200 bootstrap iterations.”

The results section has been updated as follows:

“The two clusters showed good stability across the 200 bootstrap iterations. Mean Jaccard similarity was 0.767 for cluster H and 0.796 for cluster L.”

10. The predominant genera (e.g. higher Bacteroides/Alistipes in the lower-risk cluster, higher Blautia/Clostridium in the higher-risk cluster) are interesting. The discussion could be enriched by relating these to known physiology or prior studies (for example, some Bacteroides species are linked to leaner phenotypes). This would highlight the novelty of identifying these genera in a cardiovascular risk context.

We thank the reviewer for the comments. The following paragraph in our manuscript outlines some mechanistical insights linking microbiota and cardiovascular risk:

“Additionally, Cluster L was enriched in several beneficial bacterial taxa. Notably, Bacteroides, inversely associated with aging, was the most prominent taxon favoring Cluster L [23]. Furthermore, Cluster L was enriched in Alistipes and Parabacteroides, associated with healthy aging [24], and Akkermansia, inversely correlated with obesity [25],and positively associated with healthy aging [24, 26]. Conversely, Cluster H demonstrated enrichment of bacteria associated with metabolic disease such as Blautia, Megasphaera, Fusicatenibacter, and Dorea [7, 9]. Furthermore, bacteria linked to CAD including Actinomyces, Ruminococcus gnavus, Enterococcus, Blautia, and Streptococcus were also more prevalent in Cluster H [5, 27].”

11. The manuscript is generally well-organized, but some areas could be clearer. Define the cluster labels (‘Cluster H’ and ‘Cluster L’) at first mention to avoid confusion. Consider using descriptive terms (“high-risk” vs “low-risk”) alongside the letters.

We thank the reviewer for the observation. Upon first appearance we have now named “high risk cluster” (Cluster H) and “low-risk cluster” (Cluster L) to facilitate the readers.

12. Kindly Re-run the Cox models including sex and other differing baseline factors (exercise, diet category, medication use) to confirm the cluster effect is independent. Report the full multivariable results (in Supplement) to demonstrate which covariates were significant.

We thank the reviewer for the comment. We have now added three other independent models in Supplementary Table 1, being mindful of the 10 event-per-variable rule of thumb. The new models show that the strongest predictors of mortality remain Cluster assignment and previous history of coronary artery disease.

13. The finding that clusters predicted MACE mainly in patients without established CAD is interesting. It could be explored further or at least discussed (is it due to sample size or a real difference?). If feasible, test interaction between cluster and CAD status.

We thank the reviewer for this comment. As shown in Supplementary Table 1, we tested the interaction between cluster assignment and CAD status. We did not find a significant interaction (HR = 4.11; 95% CI = 0.7-24.8; p = 0.117), suggesting that the association between microbiome clusters and MACE is not solely driven by the presence of established atherosclerotic disease. As the reviewer pointed out, the predictive value of the clusters was more apparent in patients without established CAD. We hypothesize that in individuals with existing CAD, the strong prognostic impact of atherosclerosis may overpower microbiome effects. In contrast, among patients with cardiovascular risk factors but no overt CAD, microbiome signatures linked to clinical or subclinical inflammation and metabolic dysregulation may serve as stronger predictors of future MACE. We acknowledge as well that verification by larger studies may be needed.

The observation pointed out by the reviewer is further elaborated in the manuscript as follows:

“Intriguingly, while CAD patients were more likely to belong to Cluster H, the clusters themselves did not predict MACE within the CAD subgroup. While decreased statistical power due to smaller sample size may partly explain this observation, it is also plausible that, in the context of established CAD, the influence of the GM on MACE is overshadowed by more dominant factors such as disease severity, pharmacologic interventions, and potentially limited influence of lifestyle factors at advanced disease stages. In contrast, among patients without established CAD, who are in earlier stages of CV

---

## [Decision Letter · Decision Letter 1]

4 Jan 2026

Gut microbiome compositional clusters in association with cardiovascular risk: An observational cohort study

PONE-D-25-17306R1

Dear Dr. Lerman,

We’re pleased to inform you that your manuscript has been judged scientifically suitable for publication and will be formally accepted for publication once it meets all outstanding technical requirements.

Kind regards,

Prof. Satish G Patil, PhD

Academic Editor

PLOS One

Additional Editor Comments (optional):

Reviewers' comments:

Reviewer's Responses to Questions

**Comments to the Author**

Reviewer #1: All comments have been addressed

Reviewer #2: All comments have been addressed

2. Is the manuscript technically sound, and do the data support the conclusions?

Reviewer #1: Yes

Reviewer #2: Yes

3. Has the statistical analysis been performed appropriately and rigorously?

Reviewer #1: Yes

Reviewer #2: Yes

4. Have the authors made all data underlying the findings in their manuscript fully available?

Reviewer #1: Yes

Reviewer #2: No

5. Is the manuscript presented in an intelligible fashion and written in standard English?

Reviewer #1: Yes

Reviewer #2: Yes

Reviewer #1: The term “Bacteroid” is taxonomically imprecise and does not represent a valid bacterial genus or species.

The correct and accepted term is Bacteroides, which is a recognized bacterial genus.

Using “Bacteroid” may lead to scientific ambiguity and could be flagged by reviewers or editors.

Reviewer #2: Thank you for your efforts in revising this manuscript. All comments have been carefully addressed, and the manuscript has been revised well.

**Do you want your identity to be public for this peer review?** For information about this choice, including consent withdrawal, please see our Privacy Policy

Reviewer #1: **Yes:** Hajar Fauzan Ahmad

Reviewer #2: No

---

## [Editor Report · Acceptance letter]

PONE-D-25-17306R1

PLOS One

Dear Dr. Lerman,

I'm pleased to inform you that your manuscript has been deemed suitable for publication in PLOS One. Congratulations! Your manuscript is now being handed over to our production team.

Kind regards,

on behalf of

Prof. Dr. Satish G Patil

Academic Editor

PLOS One